# A Role of Complement in the Pathogenic Sequelae of Mouse Neonatal Germinal Matrix Hemorrhage

**DOI:** 10.3390/ijms23062943

**Published:** 2022-03-09

**Authors:** Mohammed Alshareef, Khalil Mallah, Tyler Vasas, Ali Alawieh, Davis Borucki, Christine Couch, Jonathan Cutrone, Chelsea Shope, Ramin Eskandari, Stephen Tomlinson

**Affiliations:** 1Department of Neurological Surgery, Medical University of South Carolina, 301 CSB, Charleston, SC 29425, USA; alsharee@musc.edu; 2Department of Microbiology and Immunology, Medical University of South Carolina, 173 Ashley Avenue, BSB 204, MSC 504, Charleston, SC 29425, USA; mallah@musc.edu (K.M.); couchchr@musc.edu (C.C.); 3College of Medicine, Medical University of South Carolina, Charleston, SC 29425, USA; vasasj@musc.edu (T.V.); boruckid@musc.edu (D.B.); cutrone@musc.edu (J.C.); shopec@musc.edu (C.S.); 4Department of Neurological Surgery, Emory University School of Medicine, Atlanta, GA 30322, USA; ali.mostafa.alawieh@emory.edu; 5Department of Neurosciences, Medical University of South Carolina, Charleston, SC 29425, USA; 6Ralph Johnson VA Medical Center, Charleston, SC 29401, USA

**Keywords:** germinal matrix hemorrhage, hydrocephalus, neuroinflammation, complement, microglia, pediatric

## Abstract

Germinal matrix hemorrhage (GMH) is a devastating disease of infancy that results in intraventricular hemorrhage, post-hemorrhagic hydrocephalus (PHH), periventricular leukomalacia, and neurocognitive deficits. There are no curative treatments and limited surgical options. We developed and characterized a mouse model of GMH based on the injection of collagenase into the subventricular zone of post-natal pups and utilized the model to investigate the role of complement in PHH development. The site-targeted complement inhibitor CR2Crry, which binds deposited C3 complement activation products, localized specifically in the brain following its systemic administration after GMH. Compared to vehicle, CR2Crry treatment reduced PHH and lesion size, which was accompanied by decreased perilesional complement deposition, decreased astrocytosis and microgliosis, and the preservation of dendritic and neuronal density. Complement inhibition also improved survival and weight gain, and it improved motor performance and cognitive outcomes measured in adolescence. The progression to PHH, neuronal loss, and associated behavioral deficits was linked to the microglial phagocytosis of complement opsonized neurons, which was reversed with CR2Crry treatment. Thus, complement plays an important role in the pathological sequelae of GMH, and complement inhibition represents a novel therapeutic approach to reduce the disease progression of a condition for which there is currently no treatment outside of surgical intervention.

## 1. Introduction

Germinal matrix hemorrhage (GMH) is the most common neurologic pathology in neonates, which is estimated at 3.5 per 1000 live births [1]. It is caused by the disruption of fragile vasculature in the highly vascular subventricular zone (SVZ). Risk factors for GMH include pre-term birth (<32 weeks) or low weight (<1500 g), with rates of 20–40% in these infant groups [2]. Germinal matrix hemorrhage often leads to intraventricular hemorrhage (IVH), resulting in post-hemorrhagic hydrocephalus (PHH) and periventricular leukomalacia [3]. These two progressive pathological processes negatively impact neurodevelopmental processes and are highly associated with the development of cerebral palsy, with a rate of 30–42% of significant disability following severe GMH-IVH [4,5]. Furthermore, patients with severe GMH-IVH have an approximately 90% rate of associated morbidity and mortality within two years [5]. There are no medical treatments for GMH or its sequalae [6]. A currently used procedure of surgical cerebrospinal fluid (CSF) diversion can mitigate the effects of PHH, but it does not cure the neurological disability caused by progressive damage from the hemorrhage, and there is life-long surgical morbidity in up to 90% of patients, including surgical infections and shunt malfunctions [7,8]. PHH contributes to significant neurologic disability as well as visual pathway disruption and papilledema, which is a feature that can be improved with CSF diversion as well [9,10].

As in traumatic brain injury (TBI) and stroke, brain injury following GMH involves an unpredictable primary insult, which is followed by secondary injury driven, at least in part, by neuroinflammation [11]. The primary injury of GMH cannot be prevented by intervention due to the immediate and unexpected mechanical trauma of a hemorrhagic mass, which is accompanied by a surrounding ischemic insult. On the other hand, secondary injury involves a progressive injury that extends into neighboring tissue, with breakdown of the blood–brain barrier and resultant cytotoxic edema [11]. There is evidence that inflammation and increased cytokine and chemokine release in the setting of GMH contribute to the propagation of the secondary injury and are associated with subsequent periventricular leukomalacia and PHH [12,13]. A central component of inflammatory cascades is the complement system, Refs. [14,15,16], although a role for complement in GMH is unexplored. The complement system is a component of both innate and adaptive immunity, and there are three main activation pathways: the classical, lectin, and alternative pathways [17]. All pathways converge at the cleavage and activation of C3, leading to the generation of C3 opsonins (C3b, iC3b, C3d), anaphylatoxins (C3a and C5a), and finally the cytolytic membrane attack complex (MAC). Complement activation in TBI and stroke has been well studied [14,15], and complement is known to be involved in an ongoing inflammatory response that is implicated in secondary injury. In the settings of TBI and stroke, C3 opsonins can facilitate the phagocytosis of synapses and neurons, the anaphylatoxins can recruit and activate immune cells and may have both injurious and reparative roles, and the MAC can result in cell lysis, including the lysis of red blood cells that contribute to the dispersion of oxidative molecules such as heme into brain tissue and CSF [14,15,18,19].

In the current work, we describe the development and characterization of a neonatal mouse model of GMH which is based on established rat models involving the intraventricular injection of collagenase [20,21]. The murine model was optimized to have a high post-operative survivability with high rates of PHH development. We used this model for a foundational investigation into the role of complement in the development of PHH following GMH. Our data demonstrate a central role for complement in post-GMH pathology, and we show that a clinically relevant approach of complement inhibition post-GMH has a major impact on brain injury and inflammation, with subsequent benefit in terms of animal growth and longer-term motor and cognitive function. This work also establishes a direct relationship between complement activation and the development of secondary, post-hemorrhagic hydrocephalus. In this study, we used the complement inhibitor CR2Crry, a previously characterized inhibitor of C3 activation that specifically targets to sites of complement activation and C3d deposition [22]. As previously shown, localizing complement inhibition to sites of complement activation significantly increases bioavailability and efficacy, and it obviates the need to systemically inhibit the complement, thus leaving the systemic physiological functions of complement, including host-defense, intact [22,23].

## 2. Results

This paper features a germinal matrix hemorrhage model and induction of post-hemorrhagic hydrocephalus in neonatal mice. We developed a neonatal mouse GMH model with relevance to human clinical disease in several aspects, including pathophysiology of insult, post-hemorrhagic survival rate, temporal profile of pathology, rate of post-hemorrhagic hydrocephalus, and motor/cognitive delay and deficits. Surgical details of the model are in the materials and methods, and a workflow of the surgery and our subsequent complement inhibitor treatment paradigm is presented in Figure 1. The procedure resulted in a lesion and deposition of blood products in a periventricular pattern, as shown in Nissl stains of brain sections collected 24 h after injury (i.e., P5) (Figure 2a). The lesion and blood product deposition were a result of the collagenase injection and not mechanical insertion of the needle, as brain samples collected from PBS-injected animals (designated as sham) presented no lesion, blood deposition, or enlargement of the ventricles (hydrocephalus), in contrast to brains from collagenase-injected animals (Figure 2b). In addition, unlike collagenase-treated animals, neither sham animals nor naïve non-injured pups showed any denudation of the ependymal lining in the lower border of the ventricles, as shown in high-resolution Nissl-stained images (Figure 2b). Injecting collagenase at P4 resulted in a higher survival rate 24 h post injury compared to injecting collagenase at P2 or P3 (Figure 2c). A grading system from 0 to 5 was developed to characterize the severity phenotypes of brain injury and hydrocephalus, with scale 5 corresponding to global hydrocephalus. Details of the scoring system are described in the methods section, and representative images corresponding to injury scale are shown (Figure 2d). Blood products and complement deposition were evident in the perilesional hemisphere with diffuse deposition along the lesion border as depicted by immunofluorescence staining for Ter-119 (Figure 2e) and C3 (Figure 2f), thus providing justification to explore the role of complement in the context of GMH and its post-hemorrhagic sequalae. 

Targeting and tissue distribution of CR2Crry in neonates after induction of GMH. The complement inhibitor CR2Crry has been shown to bind deposited C3 activation fragments ([24]), which occurred at sites of injury in brains of collagenase-injected mice (see above, Figure 2e). To examine the targeting specificity and whole-body distribution after systemic administration of CR2Crry, we administered fluorescently labeled CR2Crry via i.p. injection after induction of GMH. Live animal fluorescence tomography showed an initial systemic distribution of CR2Crry, with subsequent localization of the drug to the brains of GMH mice but not to brains of control animals with no GMH (Figure 3). Furthermore, the quantification of fluorescence intensity revealed a CR2Crry tissue half-life of about 3 days in the brains of GMH mice, which we used as the interval between CR2Crry treatments in the therapeutic paradigm below.

Complement inhibition reduces lesion size and hydrocephalus in injured pups. To investigate the role of complement in GMH-induced pathology in a clinically relevant setting, we treated pups with CR2Crry or vehicle starting at either 1 or 24 h after collagenase injection, and every 3 days (the tissue half-life) thereafter until sacrifice at P14 (refer to Figure 1). Nissl stains of mid-hippocampal and ventricular regions from the vehicle group collected at P14 demonstrated varying degrees of parenchymal lesion along with high rates of associated intraventricular hemorrhage and PHH; no brains from the vehicle-treated group scored scale 0. On the other hand, 28% of 1 h CR2Crry-treated animals were scale 0, and 17% of 24 h CR2Crry-treated animals were scale 0. Global hydrocephalus occurred in 61% of brains from vehicle treated animals (scale 5), compared to only 7% and 22% in the CR2Crry 1 h and 24 h treatment groups, respectively (Figure 4a,b). At P45 (41 days after injury), PHH was 75% in vehicle and 33% in the CR2Crry group (*p* < 0.05) (Figure 4f).

The lesion and ventricular volumes of the experimental groups were quantified using serial Nissl stained sections through each brain. Both ventricle and lesion volumes were decreased with CR2Crry treatment (both 1 h and 24 h treatments) compared to vehicle-treated animals (Figure 4c,d). There was no significant difference between the 1 and 24 h CR2Crry treatment groups, and in subsequent experiments, we focused on 1 h CR2Crry treatment. Of note, brains from vehicle-treated animals were more likely to possess bilaterally enlarged ventricles occupying the majority of the intracranial compartment, coupled with relatively large lesions, as shown in the representative 3D reconstructed images of ventricle and lesion volume of all three conditions (Figure 4e). The lateral ventricles in brains from CR2Crry-treated animals were closer to normal ventricular anatomy compared with the visually effaced ventricles observed in the vehicle group.

CR2Crry treatment decreases perilesional complement deposition, astrocytosis, and microgliosis. We investigated the impact of complement inhibition on a perilesional cellular response in terms of post-hemorrhage astrocyte and microglia/macrophage recruitment. For analysis of astrocytosis, brain sections from P14 mice were stained for Glial Fibrillary Acidic Protein (GFAP). Astrocytosis was examined in terms of the extent of astrocytic scar extending from the lesion border inward toward intact parenchymal tissue (Figure 5a) and in terms of astrocyte density in the perilesional area at the interface with lesion (Figure 5b). Compared to vehicle-treated animals, CR2Crry-treated animals displayed reduced ipsilateral astrocytic scar formation within the surrounding brain parenchyma (Figure 5a,b). In addition, contralateral periventricular astrocytosis was also higher in vehicle animals compared to CR2Crry-treated animals (Figure 5c). Similarly, Iba-1 staining for microglia/macrophage in the perilesional region showed reduced microgliosis in CR2Crry-treated animals compared to vehicle-treated animals (Figure 5d). Correlating with reduced astrocytosis and microgliosis, there was also reduced C3 deposition in the perilesional area (Figure 5e) and ipsilateral hippocampus (Figure 5f) of CR2Crry-treated mice.

Complement inhibition results in dendritic and neuronal preservation. To explore the role of complement activation in the neurodegenerative process occurring post-hemorrhagic injury, we investigated the effect of complement inhibition on dendritic arborization (MAP2 stain) in ipsilateral and contralateral cortical hemispheres (Figure 6a). Compared to naïve mice, vehicle-treated animals displayed a decrease in dendritic arborization in both ipsi- and contralateral hemispheres, which was largely reversed with CR2Crry treatment; there was not a significant difference in MAP2 staining intensity between naïve and CR2Crry treated mice, suggesting a role for complement in dendritic loss post-injury. To further interrogate perilesional neurodegeneration, we immune-stained for neurons (NeuN). NeuN signal intensity, measured in terms of distance from lesion, was markedly higher in CR2Crry-treated mice compared to vehicle controls (Figure 6b, upper panel). Brains from CR2Crry-treated mice showed high neuronal density in the immediate perilesional space compared to the effacement of perilesional neurons in vehicle-treated mice (Figure 6b, lower panel). Notably, in vehicle-treated animals, there is a presence of non-neuronal cells in the vicinity of the lesion, as indicated by DAPI staining.

Neuronal loss is promoted by microglial/macrophage engulfment of complement opsonized neurons. We next investigated a role for microglia in complement-mediated neuroinflammation that is associated with loss of neuronal density. In Figure 5b, we analyzed perilesional neuronal density spatially extending from the lesion. Here, we analyzed overall neuronal density within perilesional fields and show that compared to naïve animals, vehicle-treated animals had a reduction in neuronal density in the perilesional area of microgliosis that colocalizes with C3 deposition. Compared to vehicle treatment, CR2Crry treatment reduced C3 deposition and microgliosis and preserved neuronal density (Figure 7a,b). To investigate whether C3 opsonization may be responsible for microglial association with neurons and subsequent neuronal loss by promoting microglia-dependent engulfment, we first quantified the colocalization of microglia/macrophages with C3-tagged neurons. Within perilesional fields of view, C3/Iba-1 colocalization was observed on 62% of NeuN+ stained cells in vehicle-treated animals compared to 20% in CR2Crry-treated animals (Figure 7c). We next demonstrated C3 deposition at the microglial/macrophage interface with neurons and quantified microglia/macrophage internalization of C3 and of neuronal (NeuN+) material. We found a higher number of microglia/macrophages with partially or fully internalized C3 in vehicle-treated animals compared to CR2Crry-treated animals (Figure 7d). From calculations using the total number of NeuN+ cells within each field as the denominator, we similarly found a higher percentage of microglia/macrophages with partially or fully internalized NeuN+ material in vehicle-treated animals compared to CR2Crry-treated animals (Figure 7e). Two examples of microglia surface interaction with and internalization of a C3-tagged neuron are shown in Figure 7f. Example 1 shows a C3-tagged neuron engulfed within a microglia/macrophage, and example 2 shows a direct interaction between a C3-tagged neuron and a microglia/macrophage (see Appendix A). These data indicate a role for complement-dependent microglial phagocytosis in neuronal loss after GMH.

Complement inhibition improves overall weight gain and animal survival. Weight gain was monitored from P2 until sacrifice at P14. Compared to vehicle-treated mice, the overall weight gain in this period was significantly improved for mice treated with CR2Crry and was similar to percent weight gain in naïve mice (Figure 8a,b). In the two days prior to injury, all groups were growing at a comparable percent weight gain. Then, 24 h after GMH induction (shown by purple arrow, Figure 8b), there was a deceleration in percent daily weight gain in both CR2Crry and vehicle animals until 4 days after injury (P8, shown by orange arrow). At that time point, CR2Crry animals began to exhibit an accelerated weight gain and approached the normal weight gain curve, as displayed by naïve animals. There was no difference between naïve and CR2Crry animal percent weight gains by P14. In a separate cohort of animals, survival was monitored for up to 41 days after collagenase-induced injury (P45). For this experiment, the same treatment paradigm used in the above studies was applied through P14, with subsequent CR2Crry or vehicle (PBS) treatments given weekly. Animal survival assessment began one day after injury (P5) to eliminate surgery-related deaths occurring within 24 h. CR2Crry group mortality plateaued at P25, while vehicle animal mortality continued to increase through P45, at which time the survival rate of CR2Crry-treated animals was 75% compared to 40% for vehicle-treated animals (Figure 8c). Within the vehicle cohort, four of 10 females and four of nine males survived to P45 with no significant difference in gender. In the CR2Crry group, four of five females and five of six males survived with no significant difference in gender.

Complement inhibition after germinal matrix hemorrhage enhances motor and cognitive performance at adolescence. An ongoing neuroinflammatory response has been linked to motor and cognitive dysfunction that is likely secondary to a loss of neurons. Our data above show an ongoing complement-dependent neuroinflammatory response and loss of neurons after GMH, and we therefore assessed whether this was linked to motor and cognitive performance at P30, when mice are able to physically perform behavioral tasks. Gait analysis (Noldus CatWalk XT) was performed at P30, and a CCI was computed using 100 plus different obtained values. Naïve and CR2Crry-treated mice had similar CCI scores, and their scores were significantly higher than vehicle-treated mice (Figure 9a). Hippocampal integrity was assessed with fear-conditioned memory retention using the passive avoidance task. CR2Crry-treated mice showed similar retention memory to naïve mice represented by a delayed time to enter the shock box of the task, which was significantly lower in vehicle-treated animals (Figure 9b). The Barnes maze task was used to assess spatial learning and memory retention, and as with the above tasks, CR2Crry-treated and naïve mice performed similarly and significantly better than vehicle-treated mice. CR2Crry-treated mice exhibited improved spatial learning ability throughout the learning phase of the task compared to vehicle-treated mice, as shown by an improved total latency on the platform and latency until first peek into the escape hole (Figure 9c). Additionally, for both latency parameters, CR2Crry treatment significantly improved animal retention memory compared to vehicle on the final day, in which animals performed the task after a 2-day break period. Heat maps depicting the movement of animals on the platform from representative experiments are shown in Figure 9d. Thus, neuroinflammation and neuronal loss after GMH correlates with behavioral deficits as mice age, and these outcomes can be reversed by complement inhibition.

## 3. Discussion

The current work utilizes the targeted complement inhibitor CR2Crry in a murine model of GMH that mimics the natural mechanism of GMH in newborn humans. Unlike previously described animal models, the clostridium-derived collagenase-based murine model described here results in a high rate of post-hemorrhagic hydrocephalus (Scale 5 lesion) in a high percentage of animals (about 60%). In comparison, human neonates with high-scale GMH (Scale 3–4) are reported to develop PHH in up to 70% of cases [25]. Autologous intraventricular blood-injection models (ABM) have also been described, but they fail to mimic the natural mechanism of GMH. Those previous models also do not induce non-traumatic germinal matrix zone vessel rupture with disruption of the SVZ, BBB, and parenchymal vasculature [26,27]. Cherian et al. described an ABM with a PHH rate of 65% following bilateral injection of autologous blood, but in the same study, they showed that the injection of artificial CSF alone caused hydrocephalus in 50% of animals, indicating that the volume of injection likely contributed to the development of hydrocephalus [26]. Other ABM studies reported significantly lower resultant PHH with about 14% success [27]. In contrast, collagenase results in vascular collagen breakdown, leading to robust neurovascular destruction that closely mimics human GMH-IVH. This mimics disruption of the BBB and the long-lasting effects of immune cell infiltration and inflammation that occurs from blood leakage into the brain tissue. The collagenase model causes neurovascular injury that also potentiates hypoxia and ischemia, as well as local immune and inflammatory responses, which may represent a limitation of this model. Nevertheless, it is of note that similar responses can be seen in human GMH pathology. Current collagenase-based models (which produced minimal to no PHH) utilized slightly older, rat, models (P7), in contrast to our mouse model (P4). Notably, the P4 induction of GMH in the mouse model equates to approximately 32-week-old premature human neonates in brain development [28], making the model directly translatable to the human pathophysiology that results in post-hemorrhagic infarction, PHH, and periventricular leukomalacia. The P7 collagenase-injected rat models equate to approximately 2.5-month-old full-term humans, in whom GMH is not encountered [21,26,29,30]. In our model, it is possible that the diffusion of collagenase or blood across the ventricle can happen from the destruction of ependymal tissue on the ipsilateral side, with leakage of active collagenase, along with blood products, through the ventricle. To this point, an advantage of our model is the trajectory used to reach the SVZ and germinal matrix. Other models use a vertical, trans-ventricular approach to reach the SVZ, whereas the current model uses a horizontal injection to penetrate the SVZ while avoiding cross-penetration of the ventricular wall, which minimizes the dispersion of collagenase through the ventricular system. This approach lowers the risk for tissue destruction resulting directly from the extravasation of collagenase, with effects more likely to occur from tissue destruction and blood dispersion into the ventricles (IVH).

GMH pathophysiology is similar to other types of brain injury, such as TBI and stroke, in which following the primary insult, there is a secondary injury phase that expands beyond that of the primary injury. With regard to TBI and stroke, complement has been shown to contribute to this secondary injury response, and complement inhibition has been shown to reduce secondary neuroinflammation and injury in experimental models [31,32]. However, the role of complement in GMH and the development of PHH has never been investigated. Multiple mechanisms have been described in the development of PHH, including iron deposition leading to inflammation and obstruction of the normal absorption pathways [33,34], TLR-4-activation leading to the hypersecretion of CSF [35], and recruitment of inflammatory cells and the formation of an astrocytic scar [21,36,37]. Although complement has been independently associated with these pathways, any direct role for complement in post-hemorrhagic hydrocephalus has not been investigated [31,38,39].

Here, we investigated the role of complement in post-GMH pathology and PHH development in a therapeutic paradigm. The complement inhibitor utilized, CR2Crry, is a fusion protein consisting of a CR2 targeting domain linked to Crry, an inhibitor of C3 activation which is a central step of the complement cascade. The CR2 moiety binds C3 activation fragments that become covalently attached to activating surfaces [40]. We initially investigated both 1 h and 24 h CR2Crry treatments post-GMH induction, since clinically delayed diagnosis of GMH is common. Treatment at both time points was protective, and there was no significant difference in outcomes between the different treatment times. Complement inhibition reduced the rate of PHH development and lesion volume, and it increased brain tissue preservation. These improvements were associated with reduced perilesional C3 deposition and reduced astrocytosis and microgliosis, the occurrence of which has been shown to contribute to the secondary injury after neurotrauma [41]. Interestingly, we identified the deposition of astrocytes in the contralateral periventricular region to be higher in vehicle compared to treated animals. This correlated with an increased rate of PHH in those animals. It is unclear whether periventricular infiltration of astrocytes contributed to hydrocephalus, but microgliosis and astrocytosis appear to be directly correlated, and both were reduced with complement inhibition. Furthermore, astrocytosis is known to play a role in minimizing the expansion of injury [42]. The reduction in astrocytosis following CR2Crry treatment in the current study may be attributed to the overall reduction in injury rather than a casual deleterious effect of astrocytes.

In addition to perilesional effects, complement inhibition reduced the deposition of C3 in the ipsilateral hippocampus and preserved dendritic density globally throughout the cortex. Clinically, high-grade GMH leads to major cognitive sequelae in up to 86% of human infants [25]. Inflammation within the hippocampus has been linked to poor neurocognitive performance in memory-related tasks, both clinically and experimentally [43]. The prevention of global hippocampal inflammation with CR2Crry likely contributed to improved Barnes maze and passive avoidance tasks performed in early adulthood testing of treated animals (P30 and beyond). Secondary to the lack of reliable fine motor and motor-related cognitive testing in younger animals, we evaluated the motor and behavioral functions of pups at P30 [44]. Our results demonstrated the preservation of motor function as well as cognitive function in CR2Crry-treated animals compared to vehicles. Maintaining larger regions of cortical tissue, both ipsilateral and contralateral to the injury site, are likely major contributors to improved motor and cognitive outcomes in CR2Crry-treated animals. CR2Crry treatment not only reduced rates of bilateral injury but also the severity of unilateral injury, with more CR2Crry-treated animals having scale 1, 2, and 3 hemorrhagic lesions relative to scale 4 and 5 in vehicle controls.

We also identified a probable mechanistic link between complement activation and neuronal loss. First, we identified a higher rate of colocalization of C3-opsonized neurons with microglia/macrophages in perilesional areas in the vehicle animals compared to CR2Crry-treated animals. Secondly, within perilesional areas, we found higher a higher percentage of microglia/macrophages with internalized C3 and neuronal material in vehicle vs. CR2Crry-treated animals. This correlated with the preservation of neuronal density in CR2Crry-treated animals that in turn was associated with improved motor and cognitive performance in adolescence. Together, the data indicate that following GMH, progression to PHH with neuronal loss and the associated behavioral deficits are mediated, at least in part, by complement receptor-mediated uptake of C3 opsonized neurons by microglia/macrophages. Although neurons appear to be phagocytosed at higher numbers in the vehicle animals, the underlying phenotype of those neurons remains unclear. Some studies have suggested that neuronal damage occurs following hemorrhagic injury due to iron-induced ferroptosis [45]. However, there is also evidence of continued, pathologic complement targeting of neurons and neuronal progenitors secondary to continued activation of the alternative pathway [14,46]. A limitation of this experimental design includes delayed gender identification at P8, which may miss potential gender differences prior to P8. Another limitation includes variability in the degree of injury to collagenase injection. This limitation was minimized by using a single surgeon to conduct injury and utilizing a pre-model training and validation with Evans blue dye.

In conclusion, PHH is a devastating pathology that is currently managed exclusively through surgical CSF diversion procedures, which carry life-long risks of repeated failure, infection, and complications [47]. Neonatal survival without surgical intervention for PHH is dismal, while those treated for hydrocephalus continue to suffer secondary brain injury-related neurological deficits, such as motor, cognitive, visual, and psychological deterioration [48]. In this study, we demonstrated a survival rate of 75% at P45 following CR2Crry treatment, independent of surgical CSF diversion, compared to 40% in vehicle-treated animals. To our knowledge, this treatment paradigm with complement inhibition is the first to demonstrate a successful preclinical pharmacologic therapy for this devastating neonatal pathology without surgery. The data suggest that complement inhibition has the potential to also reduce the rate of neonates requiring PHH-related surgical intervention. On a translational note, a humanized CR2-targeted complement inhibitor has been shown to be safe and nonimmunogenic in humans [49], and there are a multitude of companies developing complement inhibitors, with some recently approved for certain conditions [40,50].

## 4. Materials and Methods

Study design. The study design and workflow are shown in Figure 1. Animal groups in this study were Wild-type Naïve (no injury, no treatment), Sham (PBS injection in the SVZ in place of collagenase, no treatment), Vehicle (Collagenase injection in the SVZ, with intraperitoneal PBS treatment), and CR2Crry treated (Collagenase injection into the SVZ, with intraperitoneal CR2Crry treatment). Prior to the surgical procedures for GMH, animal breeders were randomly assigned to groups. Randomization was performed by an external lab personnel and was dependent on litter sizes at P1 of life in order to satisfy the numbers across groups. To minimize confounders, a single lab person performed the surgeries, treatments, testing, and scoring and was blinded to group allocations for the duration of the study. Except where indicated, surgical injection into the SVZ was conducted on post-natal day 4 (P4). The total number of animals initially randomized was n = 5 for naïve, n = 7 for sham, n = 24 for vehicle (17 survived to P5), n = 20 for CR2-Crry treatment at 1 h post-injury (14 (survived to P5), and n = 25 for CR2-Crry treatment at 24 h post-injury (17 survived to P5). Animals were excluded if they died within 24 h of surgery (<40% of animals). Study endpoints were P14 (10 days post-injury) for subacute outcome analysis, including histology, and P45 (41 days post-injury) for animal survival study and cognitive tasks. For the P45 cohort, animal gender was identified after P8, since genitalia are ambiguous prior to this time. Furthermore, the majority of deaths in the experimental groups occurred after P10, thus allowing for the capture of gender differences in survival. In the vehicle group, there were 10 females, 9 males, and 8 unknown due to death before P8. In the CR2Crry cohort, there were 5 females, 6 males, and 8 unknown due to death before P8.

Animal husbandry and care. All animal rearing, care, procedures, and euthanasia were approved by the Institutional Animal Care and Use Committee at our institution. Wild-type C57BL/J mice (Jackson Laboratory, Bar Harbor, ME, USA) were obtained at age P30 and acclimatized for 1 week. Then, animals were mated in pairs. Cages were cleaned weekly, and corn cob bedding was provided. All mice housed in the facility were exposed to 12 h light/dark cycles. Mice received access to food and water ad libitum, while pregnant females received a high-fat diet as recommended by the institutional veterinarian. All tests and experiments were conducted during the light cycle. Pregnancy and litter checks were performed daily. On the day of initial injury induction (P4), male parent mice were removed and separated from the litter. Following surgery, pups were placed on a heating pad for 30 min and then reunited with the mother. The total handling time of pups away from the mother was approximately 45 min. Then, they were monitored for an additional 60 min to ensure care of the pups by the mother. Afterwards, all animals were returned to the mouse housing facility.

Recombinant proteins and treatment paradigm. CR2Crry was prepared as previously described in [22]. Both the CR2Crry and PBS used for intraperitoneal (IP) treatment of animals were endotoxin-free. The complement inhibitory activity of the recombinant protein was verified using a zymosan assay, as previously described [22,24]. Animals in the CR2Crry treatment group were treated IP at 10 mg/kg, which was a dose previously determined to be optimal in other models [24]. Two treatment time points were used in this study: 1 and 24 h post-injury. Following the first treatment in the 1 h group, IP injections of CR2Crry were then administered at P7, P10, and P13 for a total of 4 doses. In the 24 h treatment group, the first dose of CR2Crry was given 24 h following injury, then at P7, P10, and P13. The vehicle group was treated IP with PBS 1 h post collagenase injection, then at days P7, P10, and P13. To examine tissue targeting and the tissue half-life of CR2Crry, the protein was labeled with a fluorescent marker (CF dye 92221, Biotium, Fremont, CA, USA) per the manufacturer’s protocol and administered i.p. to neonates 1 h after induction of GMH or to control animals. Live animal fluorescence tomography (Maestro II, PerkinElmer, Waltham, MA, USA) was performed at 24 h, 48 h, 72 h, 96 h, and 7 days after the single dose injection. Relative CR2Crry brain deposition was quantified by measuring signal intensity within the brain using NIH ImageJ (FIJI) integrated density.

Germinal matrix hemorrhage injury model and lesion grading system. Briefly, at post-natal day 4 (P4), mouse pups were removed from their mothers and anesthetized on ice after defining syringe insertion location. Clostridium-derived collagenase (Type VII-S collagenase, C2399, Sigma-Aldrich, St. Louis, MI, USA) was injected into the SVZ of mouse pups at P4 to induce direct spontaneous non-traumatic vessel rupture with intracerebral hemorrhage in the region of the germinal matrix and SVZ. P4 mouse pups were placed on a cooled platform to induce cryo-anesthesia as previously described [51]. A Hamilton 32-gauge needle (Model 80008, Hamilton Co., Reno, NV, USA) was used to puncture the pup’s right-sided scalp and skull with the following conditions: 1 mm posterior to the eye, 1 mm superior to the orbit, and 1 mm deep to reach the periventricular zone (Figure 1). The injection contained 0.5 units (0.5 μL) collagenase for GMH groups and PBS for the Sham group. The location of injection was chosen at the level of the periventricular region to induce parenchymal microvascular disruption (Figure 1 and Figure 2a). To optimize injections and reduce variability, the authors trained with Evans blue (EB) dye injection into the SVZ at the aforementioned parameters. Animal brains were inspected 24 h after injection to confirm the correct location of injections. At least 80% accuracy was required to perform the experiments. For this experimental design, a single researcher is recommended for all injections to maintain consistency.

Following collagenase injection, animals were placed on a heating pad for 30 min and then returned to their cages. Once the entire litter had completed the procedure and had been returned to their cage, the mother was returned to the cage and monitored for interaction with the pups. The cage was kept on the heating pad for an additional 1 h and then returned to the mouse housing facility. Sham PBS injections were performed to ensure that the hemorrhage was a result of collagenase injection and not from mechanical insertion of the needle. We confirmed that needle injection with PBS did not cause injury, while collagenase injection resulted in GMH. None of the sham group animals displayed any sign of injury at P14, and histological analysis confirmed that no lesions were present in this group. Thus, sham animals were not included as an experimental group in further experiments.

Animals were sacrificed at P14 (subacute outcomes) and P45 (chronic outcomes). In the process of developing the model, collagenase injections were also performed in P2 and P3 animals (procedural survival is shown in Figure 2c). The decision to induce injury at P4 was based on initial studies in which we found that collagenase treatment on P2 or P3 resulted in unacceptably high mortality rates (Figure 2c). Survival at 24 h after collagenase injection in pups at P2, P3, or P4 was 13%, 33%, and 65% respectively (*p* < 0.05 between P2 and P4, *p* < 0.01 between P3 and P4). Blood products identified in this model were primarily intraparenchymal on the ipsilateral (right) side. Of note, blood products were also evident along the ependymal layer of the contralateral ventricle (shown with a black arrow in Figure 2a), indicating associated intraventricular hemorrhage, which is a feature of human GMH. Using Ter-119 red blood cell (RBC) stains, we showed hematoma along the edges of the lesion three days after injury (Figure 2e). Of note, the central portions of the hematoma are washed away as an artifact of the staining process, but the RBC stain shows a layering of RBCs along the border of the lesion. The mechanism of injury in this model is the equivalent of clinical GMH grade 4 injury (intraparenchymal hemorrhage). Thus, an animal-specific injury grading system was developed to establish a distinction between parenchymal injury, ventricular involvement, and PHH (refer to Figure 2d). Scale 0 = No lesion or ventricular enlargement. Scale 1 = Lesion volume < 30% of hemispheric cortical tissue ipsilateral to injury site without ventricular involvement. Scale 2 = Lesion volume > 30% of hemispheric cortical tissue ipsilateral to injury site without ventricular involvement. Scale 3 = Lesion extending into the ipsilateral ventricle with no ventricular enlargement. Scale 4 = Lesion extending into the ipsilateral ventricle coupled with unilateral ventriculomegaly. Scale 5 = Lesion extending into both ventricles, resulting in global hydrocephalus.

Cognitive performance assessment. The Barnes maze was used to assess spatial learning and memory after GMH as previously described [52]. During the task, animals were placed on a round platform. The platform contains identical holes around the circumference, with one hole containing a safe exit. Cues were placed around the platform (triangle, square, circle) that orient the mouse to the direction of the safe exit hole. They were trained beginning at P30 for 5 consecutive days with 2 trials per day spaced 60 min apart to recognize the exit hole and enter it successfully.

Mice were given a two-day break and then re-tested using one trial for retention memory. All tasks were recorded using the Noldus EthoVision XT 13.0 system. Outcome measures included total distance traveled, latency to first poke (mouse peeking into the escape hole without entry), latency to escape hole entry, velocity, errors recorded (mouse peeking into holes other than escape hole), and time spent at different quadrants of the maze. To assess fear conditioning and learning memory, the passive avoidance task was used as previously described [53]. In brief, the test contains two chambers (one light and one dark) with an automated door between them. When the mouse enters the dark chamber, the door shuts, and a mild shock is administered. Trained mice will associate the shock with the dark chamber, thus avoiding entry. Mice were acclimatized trained at P40 and tested for memory of the shock and fear response at P44. Fear memory retention is measured by latency to enter the dark room on the test day.

Motor performance assessment. Gait analysis was performed using the automated CatWalk XT 10.6 system (Noldus Co., Leesburg, VA, USA). Mice were placed at the edge of an illuminated walkway monitored by an underlying camera. Using light scattered from contact between the animals’ paws and glass, several computed parameters for each limb were tracked, including print area, limb contact intensity (average and maximum), gait consistency, and other gait-related parameters. The average from three runs was used for trial calculations for each animal. The run duration allotted was between 0.5 and 8 s, and any run outside the given parameters was excluded. Any run that had more than 40% variation from start to finish was excluded. At P30, all experimental groups performed this task: naïve, vehicle, and CR2Crry-treated animals. Given the numerous output parameters of the device, a standardized Combined CatWalk Index (CCI) was applied as previously described [54]. In brief, parameters within each limb were assigned weighted significance, and the final CCI represents the overall performance of the animal during each run.

Tissue processing and histologic analyses. Animals in the subacute study were sacrificed at P14. Following euthanasia, cardiac perfusion was performed with cold PBS followed by 4% paraformaldehyde mixed in PBS. Then, brains were carefully extracted and placed in 4% paraformaldehyde solution overnight at 4 °C. Afterwards, the brains were moved to a new vial with 30% sucrose mixed with 4% paraformaldehyde in PBS. For tissue cutting, the brains were embedded in Tissue-Plus Optimal Cutting Temperature (OCT) compound (23-730-571, Fisher Healthcare, Waltham, MA, USA) and frozen. At time of cutting, brains were cut in 40 µm coronal sections using a freeze-mount cryostat. The complete brain was collected in 12-well plates and kept in PBS-filled wells until histologic analysis. For Nissl staining, serial brain sections 200 µm apart were mounted on a slide and stained using cresyl violet, as previously described [55]. For ventricular and lesion volume measurements, 8 serial Nissl-stained brain sections 200 µm apart and 40 µm thick were used to reconstruct the total lesion volume. Then, 4× magnification images of each slice were acquired using a Keyence BZ-X710 microscope (Keyence Co., Itasca, IL, USA). Two independent blinded observers calculated the lesion and ventricular areas using NIH ImageJ (FIJI). The average of both observers was reported. Two-dimensional (2D) analyzed images were reconstructed into 3D volumetric output files to measure brain lesion and ventricular volumes using “Free-D” software [56,57].

Immunofluorescence staining and imaging. Mid-hippocampal and mid-ventricular regions were identified by stereometric measurement using a mouse brain atlas followed by standard immunofluorescent (IF) staining as previously described [31]. All imaging and analysis were performed by a blinded lab personnel. High-resolution imaging was performed using a Zeiss LSM 880 confocal microscope (Zeiss, Carl Zeiss Microscopy, LLC, White Plains, NY, USA) at 40× zoom with water-media overlay and using the Z-stacking feature of the microscope. Images were deconvoluted using the ZEN 2.5 software (Zeiss) and reconstructed in 3D plane. Distance from the lesion edge was calculated for GFAP and NeuN analysis using ZEN 2.5 software, spectrum analysis. MAP2 arborization was calculated using spectrum analysis on ZEN 2.5 software and quantified using MATLab software (MathWorks, Inc., Natick, MA, USA). GFAP and Iba-1 perilesional signal intensity were calculated as the mean gray value (average signal intensity per pixel) using NIH ImageJ. All GFAP and Iba-1 staining was performed with negative control images (secondary antibodies only) in order to correct for underlying auto-fluorescence. Fluorescence-based analysis was performed rather than cell counting due to high cell density and clumping in the proximity of the injury site.

Colocalization analysis was performed using Imaris 9.2 (Oxford Instruments, Concord, MA, USA) for 3D image reconstruction and quantification. Neurons were quantified per field of view on Imaris 9.2. The colocalization of C3/NeuN/Iba-1 was performed by spot-to-surface interaction and reported as a percent of total neurons within the field. For the internalization of C3 or NeuN, manual quantification of partial or fully internalized particles by Iba-1 cells was quantified. The total internalized NeuN was reported as a percent of total neurons in the field. The primary antibodies used for staining were: anti-C3 (Abcam, Waltham, MA, USA, Cat. #: ab11862, 1:200), anti-NeuN (Abcam, Waltham, MA, USA, Cat. #: ab104225, 1:200), anti-Iba1 (Invitrogen, Waltham, MA, USA, Cat. #: PA5-21274, 1:80), anti-MAP2 (Abcam, Waltham, MA, USA, Cat. #: ab32454, 1:200), and anti-GFAP (Invitrogen, Waltham, MA, USA, Cat. #: 13-0300, 1:200). Secondary antibodies utilized were all donkey and include anti-rabbit Alexa Fluor 488 nm (Invitrogen, Waltham, MA, USA, Cat. #: A-21206, 1:200), anti-rat Alexa Fluor 488 nm (Invitrogen, Waltham, MA, USA, Cat. #: A-21208, 1:200), anti-rat Alexa Fluor 555 nm (Abcam, Waltham, MA, USA, Cat. #: ab150154, 1:200), anti-rabbit Alexa Fluor 555 nm (Invitrogen, Waltham, MA, USA, Cat. #: A-31572, 1:200), and anti-goat Alexa Fluor 647 nm (Invitrogen, Waltham, MA, USA, Cat. #: A32849, 1:200).

Statistical analysis. The experimental sample size was determined using power analysis and sample size estimation, performed through G*Power 3.1.9.2 tool (Franz Faul, Kiel University, Kiel, Germany). A Barnes maze performance was chosen as a reference test to calculate the effect size (estimated mean and SD). Higher or comparable effect size was also expected for the remaining tests. A calculated effect size (d) of 2.0 was anticipated when comparing GMH mice to naïve and 1.6 when comparing vehicle to CR2Crry in the treatment group based on our preliminary studies with GMH. Therefore, we used an effect size of 1.6 for our power analysis for these aims. Two-tailed analysis with a significance level α = 0.05 was considered, and then, we calculated a corrected αc = α/(number of primary comparisons) = 0.05/(2 primary comparisons) = 0.025. Ratios of group numbers were considered to be 1 (N1/N2) with an equal number of mice per group. The result of analysis reveals a sample size of 8 evaluable mice per group with an actual computed power of 84%. To ensure that a sufficient number of evaluable animals is available, we corrected for potential 40% mortality/exclusion of animals in all studies. Thus, a final number of 12 animals would be required per experimental group to satisfy the necessary minimum. Finally, in order to maintain animal litter continuity, litters were randomized into experimental groups rather than individual pups. Analyses were performed using GraphPad Prism 8.0 (GraphPad Software, San Diego, CA, USA). Parametric testing was performed unless otherwise specified in the event of a failed Brown–Forsythe test for homogeneity of variance or if normality failed. Histologic analysis for hydrocephalus was performed using Chi-squared test for ventricular volumes. Statistical analyses for lesion and ventricle sizes and IF analyses were performed using one-way ANOVA test with Bonferroni’s correction for multiple comparisons. CCI and passive avoidance tests were analyzed with a one-way ANOVA test with Bonferroni correction. Barnes maze analysis was performed with a two-way ANOVA test with Bonferroni correction. *p* values below 0.05 were considered significant. Student’s *t*-test (parametric) was used to compare two groups and was always used as two-tailed.

## Figures and Tables

**Figure 1 ijms-23-02943-f001:**
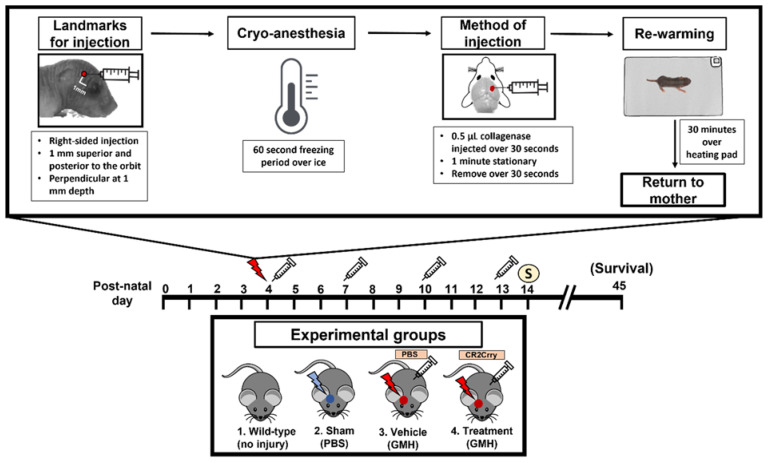
Workflow of surgical procedure and treatment paradigms for all experimental groups. At P4, the surgical coordinate was marked, and pups were cryo-anesthetized. Then, 0.5 µL of collagenase was injected into the subventricular zone of the brain via a lateral transcortical approach, and pups were returned to their mother after re-warming. The experimental groups were (1) Wild-type non-injured (Naïve), (2) PBS injected into the SVZ (Sham), (3) Collagenase injected into the SVZ and subsequent IP treatment with PBS (Vehicle control), and (4) Collagenase injected into the SVZ and subsequent IP treatment with CR2Crry. Mice received first PBS or CR2Crry treatment 1 h after collagenase injection and at subsequent 3-day intervals as indicated (depicted by syringe). Mice were euthanized for sample collection on post-natal day 14 or monitored for survival (up to day 45).

**Figure 2 ijms-23-02943-f002:**
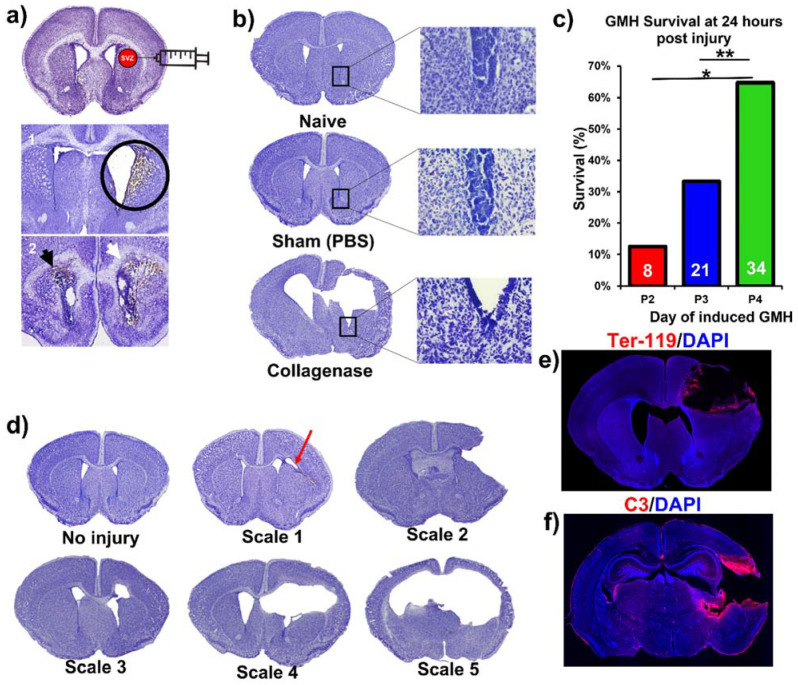
Collagenase-induced intraparenchymal GMH results in lesion coupled with hydrocephalus and hemorrhage within the SVZ. (**a**) Nissl-stained images demonstrating the collagenase injection site and showing blood product deposition within surrounding tissue in both ipsilateral and contralateral ventricles as labeled by the black and white arrowheads. (**b**) Nissl stains of naïve, sham, and vehicle brains (showing hydrocephalus), with magnified images of the lower tip of the ventricle. (**c**) Survival at 24 h after collagenase injection in pups at P2, P3, or P4. Two separate Chi-square tests were performed, one between P2 and P4, and the other between P3 and P4. * *p* < 0.05, ** *p* < 0.01. (**d**) Nissl image examples of different injury scales (0 to 5) used to categorize injury severity. (**e**) Representative IF staining of red blood cells (Ter-119 in red) and cell nuclei (DAPI in blue) in an injured P7 animal brain (3 days after injury) with deposition of blood products along the lesion. (**f**) Representative IF staining of complement deposition (C3 in red) and cell nuclei (DAPI in blue) in an injured P7 animal brain (i.e., 3 days after injury) showing extensive perilesional C3 deposition. No data points were excluded from the analysis.

**Figure 3 ijms-23-02943-f003:**
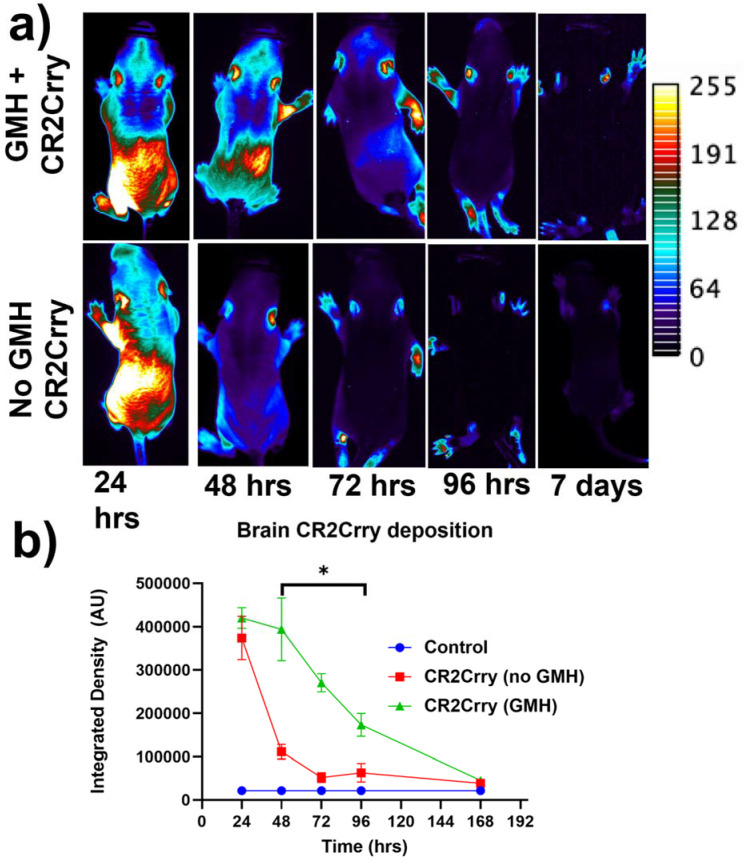
Fluorescence-tagged CR2Crry targets to the brain following injury. Fluorescent-tagged CR2Crry was administered i.p. to mice 1 hour after induction of GMH or to control mice (no injury). (**a**) Representative live animal fluorescence tomography images at indicated time points after CR2Crry administration, showing initial systemic distribution with subsequent retention of signal in the brain of GMH mice. (**b**) Quantification of fluorescence intensity in brains of GMH mice and control mice at indicated time points after CR2Crry administration, showing that the drug has a tissue half-life in the brain of about 3 days in GMH mice. Two-way ANOVA with Bonferroni’s correction for multiple comparisons. * *p* < 0.05. n = 4 for GMH, no GMH, and control groups. Error bars = mean ± SEM.

**Figure 4 ijms-23-02943-f004:**
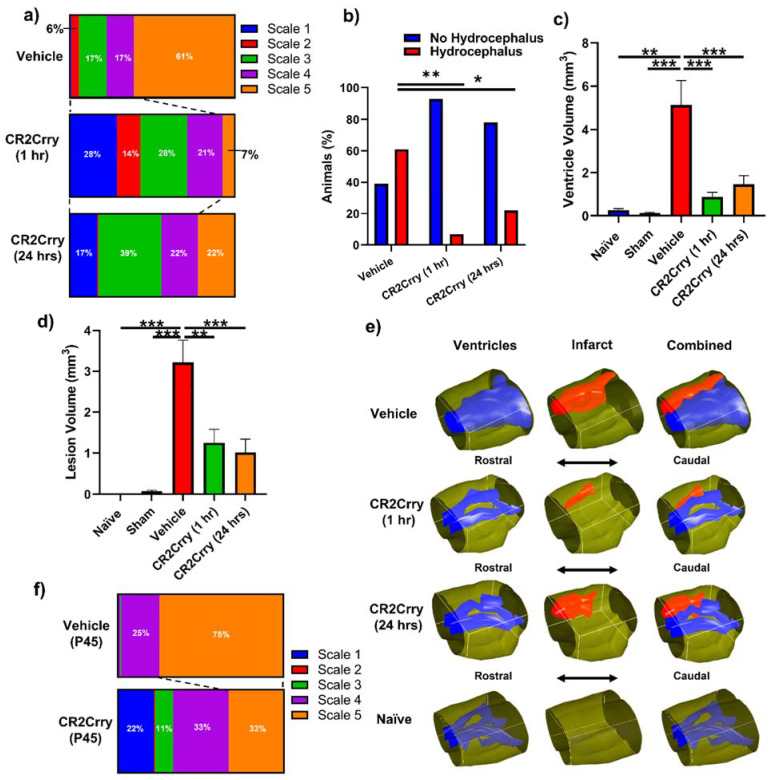
Complement inhibition leads to a reduction in lesion size and hydrocephalus. (**a**) Distribution of injury scales among the three different experimental groups: vehicle, CR2Crry (1 h), and CR2Crry (24 h). (**b**) Quantification of percent of animals that develop hydrocephalus (scale 5) vs. no hydrocephalus in vehicle, CR2Crry (1 h), CR2Crry (24 h). Chi-squared test performed between each hydrocephalus group. * *p* < 0.05, ** *p* < 0.001. (**c**,**d**) Lesion volume and ventricular volume quantification for the different groups together with naïve wild type and sham. One-way ANOVA with Bonferroni’s correction for multiple comparisons. ** *p* < 0.01, *** *p* < 0.001. n = 5 for Naïve, n = 7 for sham, n = 17 for vehicle, n = 14 for CR2Crry (1 h), and n = 17 for CR2Crry (24 h). Error bars = mean ± SEM. (**e**) Representative images with 3D reconstruction of ventricle and lesion volumes. No data points were excluded from the analysis. (**f**) Distribution of injury scales among the P45 vehicle and CR2Crry-treated groups. PHH was lower in the CR2Crry group (*p* < 0.05).

**Figure 5 ijms-23-02943-f005:**
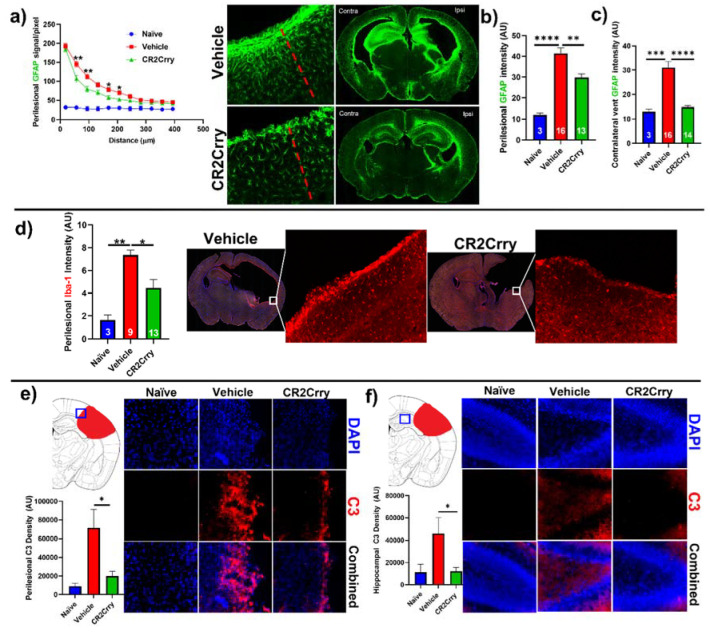
CR2Crry treatment decreases astrocyte and microglia/macrophage recruitment and decreases complement deposition. (**a**) Quantification of astrocyte signal with respect to distance in the perilesional region (left) and representative whole brain IF images (right) in CR2Crry vs. vehicle. Two-way ANOVA with Bonferroni’s correction for multiple comparisons. * *p* < 0.05, ** *p* < 0.01. n = 3 for naïve, n = 16 for vehicle, and n = 13 for CR2Crry. Error bars = mean ± SEM. (**b**) Perilesional astrocytosis. GFAP mean gray value (AU, image J) quantified along the lesion edge (within 100 µm of lesion border) showing a higher average intensity in vehicle compared to CR2Crry. Cortical mean gray value was obtained for naïve animals for comparison. One-way ANOVA with Bonferroni’s correction for multiple comparisons. ** *p* < 0.01, **** *p* < 0.0001. (**c**) GFAP mean gray value quantified for all images with visible contralateral ventricle (within 100 µm of ventricle border). There was increased periventricular astrocytosis in vehicle brains compared to both naïve and CR2Crry animals. One-way ANOVA with Bonferroni’s correction for multiple comparisons. *** *p* < 0.001, **** *p* < 0.0001. (**d**) Microglia/macrophages (Iba-1) density quantification in the perilesional area and representative images of the vehicle (left) and CR2Crry (right) treatment groups. One-way ANOVA with Bonferroni’s correction for multiple comparisons. * *p* < 0.05, ** *p* < 0.01. Error bars = mean ± SEM. (**e**) Representative images of C3 deposition with quantification of deposition in the perilesional area naïve, vehicle, and CR2Crry-treated animals, showing perilesional C3 deposition with reduction following CR2Crry treatment. One-way ANOVA with Bonferroni’s correction for multiple comparisons. * *p* < 0.05. n = 6 for naïve, n = 15 for vehicle, and n = 14 for CR2Crry. Error bars = mean ± SEM. (**f**) Representative images of C3 deposition with quantification of deposition within the ipsilateral hippocampus for naïve, vehicle, and CR2Crry-treated animals, showing perilesional C3 deposition with reduction following CR2Crry treatment. One-way ANOVA with Bonferroni’s correction for multiple comparisons. * *p* < 0.05. n = 5 for naïve, n = 12 for vehicle, and n = 14 for CR2Crry. Error bars = mean ± SEM. No data points were excluded from the analysis.

**Figure 6 ijms-23-02943-f006:**
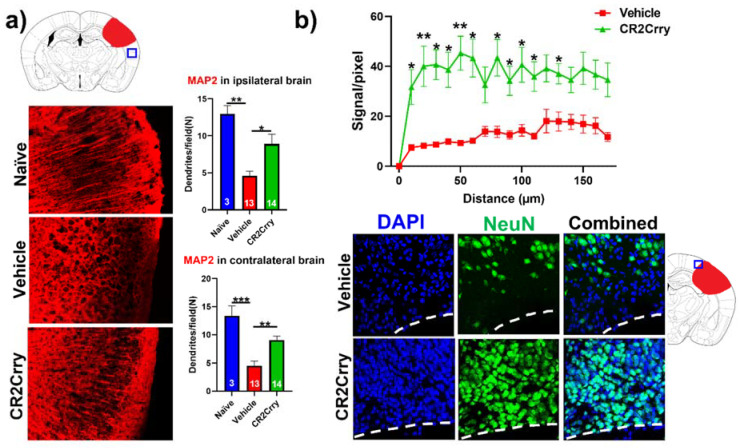
CR2Crry treatment decreases neurodegeneration by maintaining dendritic arborization and halting neuronal loss. (**a**) Representative IF images of dendritic processes (MAP2) of cortical tissue in the ipsilateral hemisphere and quantification of MAP2 dendritic density in both the ipsilateral and contralateral hemisphere of naïve, vehicle, and CR2Crry animals. One-way ANOVA with Bonferroni’s correction for multiple comparisons. * *p* < 0.05, ** *p* < 0.01, *** *p* < 0.001. Error bars = mean ± SEM. (**b**) Representative IF images of NeuN-stained sections of the perilesional area, and quantification of NeuN signal starting from lesion border inward toward the ipsilateral cortical tissue. Two-way ANOVA with Bonferroni’s correction for multiple comparisons. * *p* < 0.05, ** *p* < 0.01. n = 15 for vehicle, and n = 14 for CR2Crry. Error bars = mean ± SEM. No data points were excluded from the analysis.

**Figure 7 ijms-23-02943-f007:**
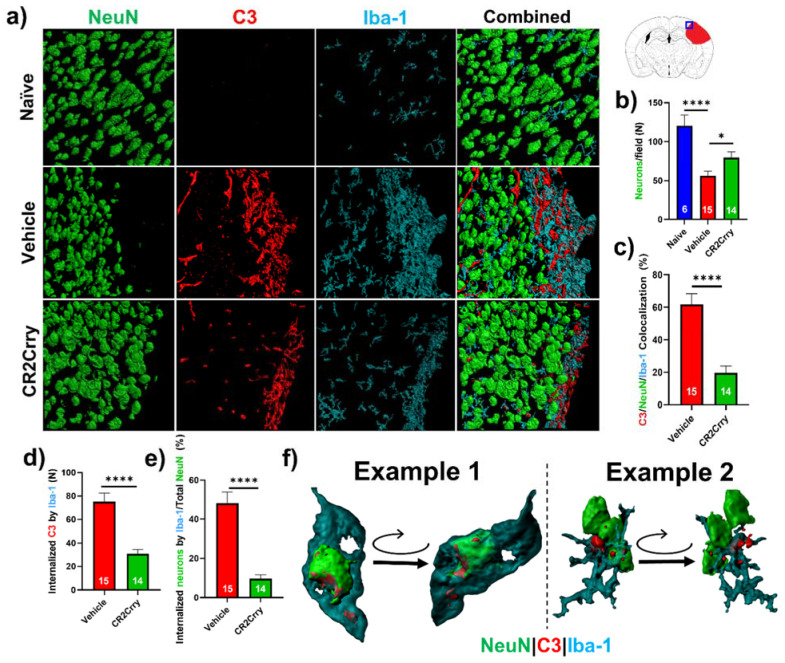
Complement inhibition reduces microglia/macrophage association with and internalization of C3-tagged neurons. (**a**) Representative images of perilesional area showing staining for Iba-1, C3, and NeuN, which were obtained using confocal microscopy with Z-stacking. Three-dimensional (3D) image reconstruction was performed using Imaris to obtain colocalization analysis. (**b**) Perilesional quantification of neurons in animals treated as indicated. Cortical images from naïve brains were used for comparison. * *p* < 0.05, **** *p* < 0.0001. Error bars = mean ± SEM. (**c**) IF stain for C3 deposition, Iba-1, and NeuN colocalization, performed based on surface proximity and analyzed as a percent of total neurons/field. Association of microglia/macrophages (Iba-1) with complement-tagged neurons as a percentage of total neurons present was higher in vehicle (62%) compared to CR2Crry-treated animals (20%). **** *p* < 0.0001. Error bars = mean ± SEM. n = 6 for naïve, n = 15 for vehicle, and n = 14 for CR2Crry. (**d**) Quantification of microglia/macrophages within the perilesional space with partial or complete internalization of C3 material. (**e**) Quantification of microglia/macrophages within the perilesional space with partial or complete internalization of NeuN+ material as a percent of total quantified neurons. **** *p* < 0.0001. Error bars = mean ± SEM. n = 15 for vehicle, and n = 14 for CR2Crry. (**f**) Example images showing microglia/macrophage association and internalization of C3-tagged neurons in vehicle animals (see also Appendix A). No data points were excluded from the analysis.

**Figure 8 ijms-23-02943-f008:**
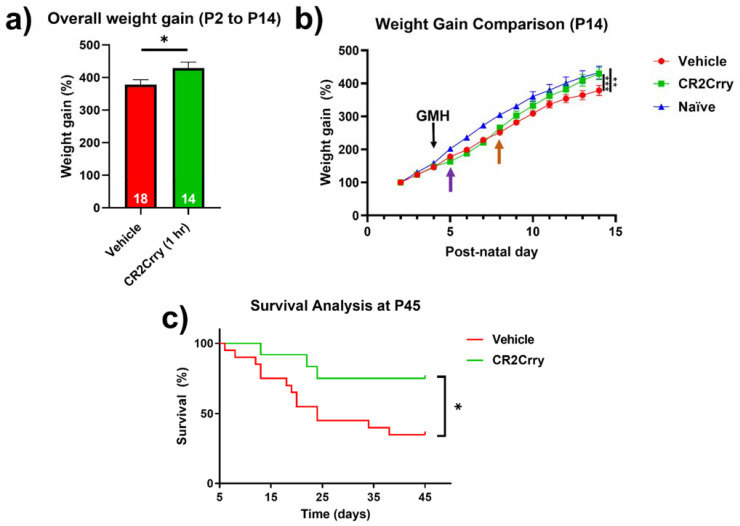
CR2Crry treatment improves weight gain after GMH and promotes survival. (**a**) The overall percent weight gain from day P2 to P14 in vehicle and CR2rry-treated mice. Unpaired Student’s *t*-test. * *p* < 0.05. Error bars = mean ± SEM. (**b**) Daily weight percent gain over a 12-day period following GMH induction. Deceleration in percent daily weight gain in CR2Crry and vehicle animals at P5 (purple arrow). Acceleration of weight gain in CR2Crry-treated animals (orange arrow). Two-way ANOVA with Bonferroni’s correction for multiple comparisons. ** *p* < 0.01, *** *p* < 0.001. n = 11 for naïve, n = 18 for vehicle, and n = 14 for CR2Crry. Error bars = mean ± SEM. (**c**) Survival assessed over 41 days after injury (P45). Animal survival was assessed beginning at one day after injury (P5). Animals that died within 24 h of injury were excluded from analysis. CR2Crry group deaths plateau around P25, while vehicle animal deaths continue until close to P45. P45 animal survival was 75% in the CR2Crry group compared to 45% in the vehicle (*p* < 0.05). Log-rank (Mantel–Cox) test. * *p* < 0.05. n = 20 for vehicle and n = 12 for CR2Crry. Error bars = mean ± SEM.

**Figure 9 ijms-23-02943-f009:**
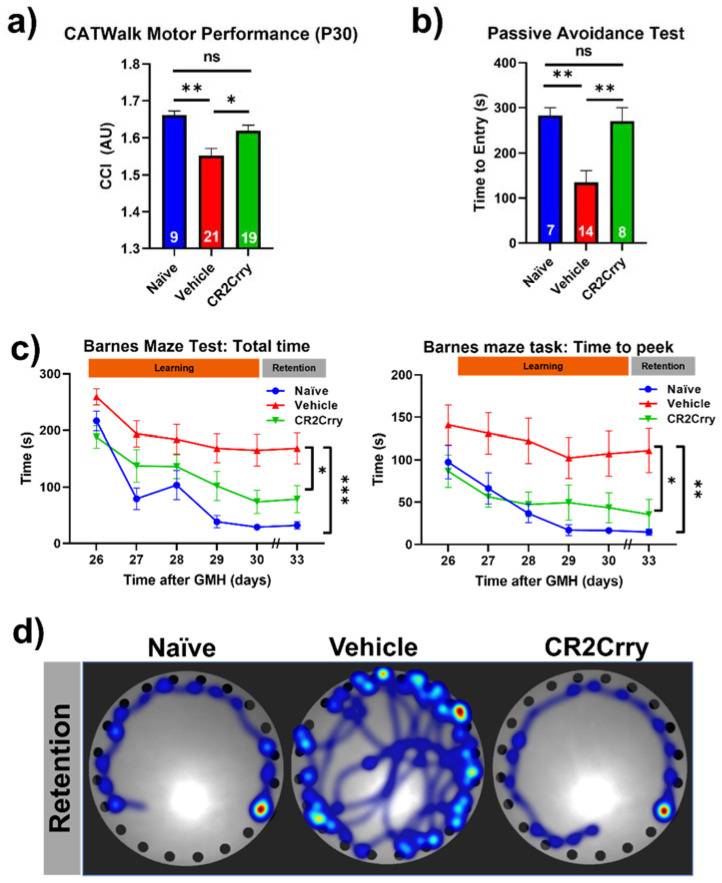
CR2Crry treatment improves motor performance and cognitive performance of GMH injured mice. (**a**) Gait analysis evaluated using Catwalk XT. The output variables for all four limbs were combined into a previously described Combined Catwalk Index (CCI) 26 days after injury (P30). One-way ANOVA with Bonferroni’s correction for multiple comparisons. * *p* < 0.05, ** *p* < 0.01. Error bars = mean ± SEM. (**b**) Passive avoidance test (time to entry) performed 36 days after injury (P40) shows improvement in fear-conditioned learning in CR2Crry animals and naïve as compared to vehicle. One-way ANOVA with Bonferroni’s correction for multiple comparisons. ** *p* < 0.01. Error bars = mean ± SEM. (**c**) Barnes maze task performed beginning at 26 days after injury (P30). There was a training period of 5 days followed by a two-day rest period and subsequently underwent testing to evaluate retention memory 33 days after injury (P37). Variables presented were total latency and latency to first peek into the escape hole. Two-way ANOVA with Bonferroni’s correction for multiple comparisons. * *p* < 0.05, ** *p* < 0.01, *** *p* < 0.001. n = 7 for naïve, n = 21 for vehicle, and n = 16 for CR2Crry. Error bars= mean ± SEM. (**d**) Representative heat maps of movement of mice from the three treatment groups on the platform on the retention day. No data points were excluded from the analysis.

## Data Availability

The data presented in this study are available on request from the corresponding author, Dr. Stephen Tomlinson.

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
