# Peer review of "A Role of Complement in the Pathogenic Sequelae of Mouse Neonatal Germinal Matrix Hemorrhage"

_ijms, 2022, doi:10.3390/ijms23062943_

Round 1

Reviewer 1 Report

The manuscript describes a series of experiments demonstrating a link between complement and outcome of GMH in a rationally designed murine model. 

Several changes in the methods section and data presentation are needed :

  1. The total number of animals  and the number randomized to each group needs to be mentioned in the study design section.
  2.  Power considerations leading to the choice of n/group need to be mentioned
  3. The sham group described in methods is not featured in any of the figures, which only feature naive/vehicle / treatment. It is not clear if this group was used in any of the assays. This is a rather important control so if it was abandoned some explanation is due.
  4. The question on of sex as a biological variable is poorly addressed and it behooves the authors to explain why they chose to determine sex in only one cohort and after p8. this needs to be explained in methods and acknowledged as a caveat/limitation in discussion, especially in view of the kown large sex differences in immune function. 

Author Response

We are resubmitting a revised version of “A Role of Complement in the Pathogenic Sequelae of Mouse Neonatal Germinal Matrix Hemorrhage” by Alsshareef et al for your consideration for publication on IJMS.

A point-by-point response to reviewer's no 1  comments is below. Changes and additions to the revised text are indicated by red font.

Reviewer 1:

The manuscript describes a series of experiments demonstrating a link between complement and outcome of GMH in a rationally designed murine model. 

Several changes in the methods section and data presentation are needed:

  1. The total number of animals and the number randomized to each group needs to be mentioned in the study design section.

Response:

Below are the total initial numbers randomized to each group, and the number of animals that survived past P5.

n=5 for naïve, n=7 for sham, n=24 for vehicle (17 survived to P5), n=20 for CR2-Crry treatment at 1 hr post-injury (14 (survived to P5), and n=25 for CR2-Crry treatment at 24 hr post-injury (17 survived to P5).

We have revised the study design section of the materials and methods to include the above information.

  1. Power considerations leading to the choice of n/group need to be mentioned

Response:

We discuss our power analysis method and choice of n/group in the statistical analysis section of the materials and methods.

“Experimental sample size was determined using Power analysis and sample size estimation, performed through G*Power 3.1.9.2 tool (Franz Faul, Kiel University, Germany). Barnes maze performance was chosen as a reference test to calculate the effect size (Estimated mean and SD). Higher or comparable effect size was also expected for the remaining tests. A calculated effect size (d) of 2.0 was anticipated when comparing GMH mice to naïve and 1.6 when comparing vehicle to CR2Crry in the treatment group based on our preliminary studies with GMH. Therefore, we used an effect size of 1.6 for our power analysis for these aims. Two-tailed analysis with significance level α = 0.05 was considered and then a corrected αc = α/(number of primary comparisons)=0.05/(2 primary comparisons)= 0.025. Ratios of group numbers was considered to be 1 (N1/N2) with equal number of mice per group. The result of analysis reveals a sample size of 8 evaluable mice per group with an actual computed power of 84%. To ensure sufficient number of evaluable animals is available, we corrected for potential 40% mortality/exclusion of animals in all studies. Thus, a final number of 12 animals would be required per experimental group to satisfy the necessary minimum. Finally, in order to maintain animal litter continuity, litters were randomized into experimental groups rather than individual pups.”

  1. The sham group described in methods is not featured in any of the figures, which only feature naive/vehicle/treatment. It is not clear if this group was used in any of the assays. This is a rather important control so if it was abandoned some explanation is due.

Response:

The sham group was included during GMH model development in order to confirm that a needle insertion alone, with injection of PBS and not collagenase, did not contribute to injury. This sham group was compared to the vehicle group in which an injection of collagenase was delivered by the same caliber needle (followed by PBS treatment). Once this step was validated, as depicted in figure 2b, we did not consider it necessary to include a sham group for the rest of the studies.

The following revised statement is included the methods section under the heading “Germinal matrix hemorrhage injury model and lesion grading system”:

“We confirmed that needle injection with PBS did not cause injury, while collagenase injection resulted in GMH. None of the sham group animals displayed any sign of injury at P14, and histological analysis confirmed that no lesions were present in this group. Thus, sham animals were not included as an experimental group in further experiments”

  1. The question on of sex as a biological variable is poorly addressed, and it behooves the authors to explain why they chose to determine sex in only one cohort and after p8. this needs to be explained in methods and acknowledged as a caveat/limitation in discussion, especially in view of the known large sex differences in immune function. 

Response:

Identifying gender in neonate mice is difficult prior to P8 as the genitalia are ambiguous. Therefore, we elected to follow animals past P8 for gender identification. Furthermore, in our P45 groups, most deaths occurred after P10, thus survival analysis was performed taking account of gender. There was no significant difference between male and female cohorts. This is detailed in the last 4 sentences of the results section “Complement inhibition improves overall weight gain and animal survival.”

We include a statement in the Materials and Methods section under the heading “study design” explaining why P8 was selected.

“For the P45 cohort, animal gender was identified after P8, since genitalia are ambiguous prior to this time. Furthermore, the majority of deaths in the experimental groups occurred after P10, thus allowing for capture of gender differences in survival”.

We also now included this as one of the limitations of our study in the discussion. The following sentence was added:

“A limitation of this experimental design includes delayed gender identification at P8, which may miss potential gender differences prior to P8.”

Reviewer 2 Report

This topic sounds interesting and paper is well written. Please look at these points:

  • Lines 101-107: This part seems Methods and not results. Please revise.
  • Lines 356-359: Discuss about the role og hydrocephalus, look at these two important refs:  -- Resolution of Papilledema Following Ventriculoperitoneal Shunt or Endoscopic Third Ventriculostomy for Obstructive Hydrocephalus: A Pilot Study. Medicina. 2022; 58(2):281. doi: 10.3390/medicina58020281 --- Papilledema in children with hydrocephalus: incidence and associated factors. J Neurosurg Pediatr. 2017 Jun;19(6):627-631. doi: 10.3171/2017.2.PEDS16561. 
  • Line 386-389: "Complement activation has been implicated in propagating secondary injury following TBI and stroke" What do authors mens ? Please improve this point.
  • Lines 421-423 "Prevention of global hippocampal inflammation with CR2Crry likely contributed to improved Barnes maze and passive avoidance tasks performed in early adulthood testing of treated animals (P30 and beyond)" How these tests were performed? Please report this in the text.
  • Does this paper have any limitations ? small sample? Please report.
  • Lines 561 "one between P2 and P4, and the other between P3 and P4. *p<0.05, **p<0.01." Was this report in the text?
  • it could be better to put Methods section before Results section.

Author Response

We are resubmitting a revised version of “A Role of Complement in the Pathogenic Sequelae of Mouse Neonatal Germinal Matrix Hemorrhage” by Alsshareef et al for your consideration for publication on IJMS.

A point-by-point response to reviewer 2 comments is below. Changes and additions to the revised text are indicated by red font.

Reviewer 2:

This topic sounds interesting and paper is well written. Please look at these points:

  • Lines 101-107: This part seems Methods and not results. Please revise.

Response:

We have rearranged a significant part of this section and in accordance with this comment have moved portions of the text to the material and methods section under “Germinal matrix hemorrhage injury model and lesion grading system”.

  • Lines 356-359: Discuss about the role of hydrocephalus, look at these two important refs:  -- Resolution of Papilledema Following Ventriculoperitoneal Shunt or Endoscopic Third Ventriculostomy for Obstructive Hydrocephalus: A Pilot Study. Medicina. 2022; 58(2):281. doi: 10.3390/medicina58020281 --- Papilledema in children with hydrocephalus: incidence and associated factors. J Neurosurg Pediatr. 2017 Jun;19(6):627-631. doi: 10.3171/2017.2.PEDS16561. 

Response:

We now include a statement in the introduction on the effect of PHH on neurocognitive deficits and on visual pathway decline with papilledema. We also note that papilledema does indeed improve with CSF diversion, as noted in the above references.

Line 386-389: "Complement activation has been implicated in propagating secondary injury following TBI and stroke" What do authors mean? Please improve this point.

Response:

Similar to TBI and Stroke, GMH includes a primary injury phase and a secondary injury phase which involves a neuroinflammatory response. We and others have previously shown that complement propagates the secondary injury phase in TBI and stroke, and for that reason we hypothesized that complement is also implicated in GMH.

We have revised the statement to improve clarity, as follows:

“GMH pathophysiology is similar to other types of brain injury, such as TBI and stroke, in which following the primary insult there is a secondary injury phase that expands beyond that of the primary injury. With regard to TBI and stroke, complement has been shown to contribute to this secondary injury response, and complement inhibition has been shown to reduce secondary neuroinflammation and injury in experimental models”.

  • Lines 421-423 "Prevention of global hippocampal inflammation with CR2Crry likely contributed to improved Barnes maze and passive avoidance tasks performed in early adulthood testing of treated animals (P30 and beyond)" How these tests were performed? Please report this in the text.

Response:

We have expanded on both the barnes maze and passive avoidance tasks in order to explain the platforms and expected outcomes.

Please see the Methods section, subsection: Cognitive performance assessment.

“During the task, animals were placed on a round platform. The round platform contains identical holes around the circumference, with one hole containing the safe exit. Cues were inserted surrouning the platform (triangle, square, circle) that orient the mouse to the direction of the safe exit hole. They were trained beginning at P30 for 5 consecutive days with 2 trials per day spaced 60 minutes apart to recognize the exit hole and enter it successfully”.

And…

“In brief, the test contains two chambers (one light and one dark) with an automated door between them. When the mouse enters the dark room, the door shuts and a mild shock is administered. Trained mice will recognize the shock and associate it with the dark room, thus avoiding entry. Mice were acclimatized trained at P40 and tested for memory of the shock and fear response at P44. Fear memory retention is measured by latency to enter the dark room on the test day”.

  • Does this paper have any limitations ? small sample? Please report.

Response:

Yes, one limitation includes not identifying gender prior to P8. Another is the variability in animal response to injury, requiring higher sample sizes in order to see trends. These are now included in the discussion section:

“Limitations of this experimental design include gender identification at P8, thus missing potential gender differences prior to P8. Another limitation includes variability in the degree of injury to collagenase injection. This limitation was minimized by using a single surgeon to conduct injury, and utilizing a pre-model training and validation with Evans blue dye”.

In addition, please see response to reviewer no. 1, point no. 4 in which we address limitation of delayed gender identification.

  • Lines 561 "one between P2 and P4, and the other between P3 and P4. *p<0.05, **p<0.01." Was this report in the text?

Response:

This is now included in the methods section, subsection: Germinal matrix hemorrhage injury model and lesion grading system:

“The decision to induce injury at P4 was based on initial studies in which we found that collagenase treatment on P2 or P3 resulted in unacceptably high mortality rates (Fig. 2c). Survival at 24 hours after collagenase injection in pups at P2, P3, or P4 was 13%, 33%, and 65% respectively (p<0.05 between P2 and P4, p<0.01 between P3 and P4).”

  • it could be better to put Methods section before Results section.

Response: According to the author instructions for IJMS, the following order should be used: Introduction, Results, Discussion, Materials and Methods, and Conclusions (Optional).

Round 2

Reviewer 2 Report

Authors solved all my criticisms